# Updating Distribution, Ecology, and Hotspots for Three Amphibian Species to Set Conservation Priorities in a European Glacial Refugium

**Ilaria Bernabò** [1] , **Viviana Cittadino** [1] , **Sandro Tripepi** [1] , **Vittoria Marchianò** [2] , **Sandro Piazzini** [3] , **Maurizio Biondi** [4] and **Mattia Iannella** [4,*]

1 Department of Biology, Ecology and Earth Science, University of Calabria, Via P. Bucci 4/B, Rende, 87036 Cosenza, Italy
2 Pollino National Park, 85048 Rotonda, Italy
3 Department of Physical Sciences, Earth and Environment, University of Siena, Via P.A. Mattioli 4, 53100 Siena, Italy
4 Department of Life, Health & Environmental Sciences, University of L'Aquila, Via Vetoio-Coppito, 67100 L'Aquila, Italy
* Correspondence: mattia.iannella@univaq.it

**Abstract:** The Calabrian Peninsula (Southern Italy) has acted as a glacial refugium and is now considered a hotspot for the genetic diversity of several species. Even if it hosts the highest diversity of many Italian endemic amphibian species, the distribution of some of these needs an update to address conservation measures. We took advantage of a vast dataset for three Italian species (*Bombina pachypus*, *Salamandrina terdigitata*, *Triturus carnifex*), two of which are endemic, deriving from a 40-year field surveys dataset (1982–2022), to update their distribution and basic ecological requirements. We evaluated changes in their distribution, projecting them on a broader spatial scale through a kernel density estimation, inferring statistically-significant hotspots using Corine Land Cover patches, and assessing the protected areas' coverage. We confirmed that Pollino, Catena Costiera, Sila and Aspromonte massifs are the main statistically-significant hotspots. Kernel densities showed a diversified pattern of gains/losses, sometimes overlapping, depending on the species. The whole outcomes obtained allow us to pinpoint specific areas where effective conservation measures need to be applied. Ousr findings reveal that local-scale monitoring and management should be planned, especially within the existing nationally-designated protected areas, which have been shown to protect far less with respect to the Natura 2000 sites.

**Keywords:** amphibian distribution; hotspot analysis; protected areas; conservation; *Bombina pachypus*; *Salamandrina terdigitata*; *Triturus carnifex*; Italian Peninsula; Calabria; Southern Basilicata

## 1. Introduction

Global fluctuations in climate during the Quaternary produced significant changes in the geographic distribution of the populations and species genomes of the European fauna [1]. Mediterranean regions are renowned as biodiversity hotspots, in which several taxa survived the Pleistocene ice ages in refugia and then expanded [1,2]. Italy, also due to its complex history, has had its own main refugia in the southernmost part of the Peninsula, encompassing the Calabrian region, which currently harbours the highest genetic diversity for several taxa [2–10].

Peninsular Italy and its surrounding islands are a hotspot for the herpetofauna; in particular, amphibians count for the highest level of diversity and endemism, including about half of the species in Europe [11,12]. However, this high richness in biodiversity and endemic lineages is exposed to anthropogenic changes, facing both a local and global decline. In Italy and Europe, respectively, 36% [13] and 23% [11] of amphibian species are included in threatened categories,

as also observed for the highest rate of the global decline of amphibians, which represents the archetype example of the current biodiversity crisis [14,15]. Concerning Italy, the observed reduction of amphibian fauna stems from the general loss of suitable habitats to specific impacts on breeding wetlands due to multiple factors (land-use change, pollution, water drainage caused mainly by agricultural intensification, fragmentation, introduction of invasive alien species, and, in the case of artificial aquatic habitats, abandonment of traditional management) [16–21]. Moreover, future scenarios predict dramatic impacts on Italian amphibians relating to climate change and the spread of emerging amphibian diseases [22–25].

The Habitats Directive (92/43/EEC), together with the Birds Directive, represents the cornerstone of the European Union's nature and biodiversity policy, establishing the Natura 2000 project, a network of sites designed to ensure the conservation of Europe's most valuable and threatened species and habitats. Numerous rare, threatened, or endemic Italian amphibian species are included in Annexes II, IV and V of the Habitats Directive (for a total of 34 out of 44 occurring in Italy). In the last overall assessment (2013–2018) of conservation status and trends for the species of community interest, about 30% of the amphibian species shows poor or bad status and close to 50% have negative trends [26]. At the Italian level, data from monitoring and reporting showed a critical general framework for amphibians and the need for knowledge improvement on population status and trends on the local scale [27].

This need is particularly urgent in the southernmost portion of the Italian Peninsula, where amphibians' current distribution and conservation status is partially known or under-investigated. The available studies for the south are dated, scattered, or limited to a few species. Notwithstanding their significant importance at the biogeographical, conservation and European Union levels, little updated information exists for three amphibians listed in Annex II of the Habitats Directive (available at: https://eur-lex.europa.eu/legal-content/EN/TXT/?uri=CELEX:31992L0043, accessed on 15 June 2022): *Bombina pachypus*, *Salamandrina terdigitata* and *Triturus carnifex*.

Our study focused on these three species in the Calabrian Peninsula and the surrounding territories, an area recognized as a glacial refugium and hotspot of genetic diversity for many species, as stated above.

Here, we aim to expand and update spatial information about the three species, using information from past data, with the most recent obtained through multi-year field surveys. Specifically, incorporating observations collected over 40 years, our objectives are: (1) to compare the historical and current distribution of selected species at the regional level; (2) to shed light on the species' ecological requirements and habitats (3) to define the magnitude of occurrence changes across the region and thus identify areas for in-depth further investigations and conservation actions; (4) to quantify per-species territories' loss and gains in the network of Protected Areas; and concurrently (5) to infer statistically-significant hotspots to support management actions.

## 2. Materials and Methods

### 2.1. Target Species and Study Area

Considering their biogeographic and conservation importance, we selected three target species, namely the Apennine yellow-bellied toad, *Bombina pachypus* (Bonaparte, 1838), the Southern spectacled salamander, *Salamandrina terdigitata* (Bonnaterre, 1789), and the Italian crested newt, *Triturus carnifex* (Laurenti, 1768). The target species are listed in Appendix II of the Bern Convention and Annexes II and IV of the Habitats Directive.

*Bombina pachypus* is endemic to Italy, and its range spans from the Northern to Southern Apennines in peninsular Italy [28]. There is currently no full agreement on the taxonomic status of *Bombina pachypus*: some authors consider it as subspecies of *B. variegata* (Linnaeus, 1758) based on molecular and phylogenetic evidence [29–32], whereas others recognize Appennine yellow-bellied toads as a valid species considering morphological and genetic differences [3,33,34]. In the present work, we considered *B. pachypus* a distinct species. This anuran inhabits ephemeral shallow ponds, artificial water bodies, and temporary

small streams in forests and open areas [35,36]. A significant decreasing population trend was observed for this anuran species during recent decades, so *B. pachypus* is currently considered "Endangered" by the IUCN [13]. Recent research even predicts a possible future worsening of the situation in some areas within its range [37].

*Salamandrina terdigitata* is an endemic species of southern Italy (from the south of Campania to the southern tip of Calabria), separated based on molecular and morphological evidence from *S. perspicillata* [38–40]. In this forest-dwelling species, only females are aquatic during the oviposition phase; this species prefers lotic, well-oxygenated water with predominantly rocky beds, but it also reproduces in artificial habitats [41,42]. *S. terdigitata* is currently listed as of "Least Concern", even though an update is reported to be needed [42].

In Italy, *Triturus carnifex* is distributed in continental (marginally in the Valle d'Aosta and Trentino-Alto Adige) and peninsular regions and with the southern limit in central Calabria [28,43,44]. This newt occurs in various natural or semi-natural terrestrial environments and is frequently found in permanent and temporary aquatic habitats during the reproductive period [44]. It is listed as "Near Threatened", and a decreasing population trend is reported by the Red List of Italian Vertebrates [13]; the causes of this decline are attributed to intensive farming and pollution, habitat loss, and the introduction of aquatic predators [16,20–22,45].

To define the study area encompassing as much biogeographic, genetic, and conservation significance as possible, we followed the geographic patterns published for the three target species. As stated above, the Calabrian Peninsula and the surrounding areas are a crucial hotspot of diversity [1,5,46]. Thus, we merged spatial data for *B. pachypus* [3,37], *S. terdigitata* [6,47], and *T. carnifex* [46], encompassing the areas where considerable genetic diversity is hosted in terms of haplogroups and excluding the territories where sympatry or introgression are reported (e.g., [48]). Our study area is therefore represented by the Calabrian Peninsula and a part of southern Basilicata, which comprises the Pollino National Park, the largest of the Italian protected areas (1926 km$^2$). The Pollino massif marks the border between Calabria and Basilicata. It defines an almost continuous mountain system from the Ionian Sea to the Tyrrhenian Sea with elevations ranging from 52 to 2267 m a.s.l. This area exhibits a very complex landscape characterized by deep gorges produced through erosion by rivers and streams; following an altitudinal gradient, dominant vegetation ranges from the Mediterranean to Alpine habitat types with *Pinus heldreichii* subsp. *leucodermis*, which represents the particular entity of the park (see for details Pignatti et al. [49]). Calabria is the extremity of the Italian Peninsula, covering an area of 15,080 km$^2$. It is a predominantly hilly and mountainous region, crisscrossed by the Apennines and with limited plain areas. The bioclimate is typically Mediterranean with seasonal oceanic rainfall up to about 1000 m, and temperate sub-Mediterranean in the mountainous area [50]. The complex geological setting and bioclimatic diversity generate a considerable variety in vegetation [51,52]. Calabria is among the regions with the highest forest cover in Italy [53]. Briefly, the forest vegetation consists of evergreen sclerophyllous formations of *Quercetea ilicis* up to 600–800 m a.s.l. and deciduous winter formations of *Querco-Fagetea* in the mountain zone up to 2000 m a.s.l. with *Pinus nigra* subsp. *laricio*, *Fagus sylvatica*, and *Abies alba*. The typical species of the Mediterranean maquis, such as *Olea europaea* var. *oleaster*, *Quercus suber* and *Myrtus communis*, are distributed in the coastal zone [52]. In coastal areas, wetlands have been significantly reduced over the last century due to land reclamation and anthropogenic activities, regulation of watercourses and urbanization [54,55].

*2.2. Data Collection*

Exact localization using GPS has been used to collect the target species' occurrence localities. We classified them into historical, recent, and confirmed data (i.e., localities where the species still occurs, from historical records to the recent field surveys). Specifically, our herpetological database (DiBEST, University of Calabria; Calabrian section of SHI, and Dipartimento Ambiente, Regione Calabria) has been gathered from several monitoring projects and, where available, technical reports from protected areas. It contains georeferenced records from two long-term datasets: historical data covering observations from

1982 to 2006, for a total of 438 occurrence sites, partially published in Sindaco et al. [28], Sperone et al. [56,57]; Talarico et al. [35]; Tripepi et al. [58], and recent data deriving from field research conducted from 2013 to 2022. All these data, as well as their sources, are reported in Supplementary Table S1. Before the recent extensive field surveys, potential aquatic habitats were identified by cartographic recognition on the maps produced by the Istituto Geografico Militare (IGM, 1:25,000) and by satellite images. Sampling efforts were mainly focused from late winter to late summer, during the period of activity of the three species according to their phenology. The traditional methodologies for the detection and characterization of amphibians were used [59,60]. Briefly, we assessed the presence of amphibian species by visual encounter surveys for adults, egg clutches and larvae, by listening to the calls of adult males for *B. pachypus*, and by dip-nettings in aquatic sites. *S. terdigitata* were sampled by searching beneath logs, rocks, leaf–litter, trees' buttresses, and cavities. Since investigation covers a large part of the Calabria region and southern Basilicata, we visited, with a minimum of two visits, a subsample of locations of historical occurrence in the Pollino, Sila, Serre and Aspromonte massif and along the Catena Costiera.

Field surveys involved both aquatic artificial and natural habitats that we categorized in the following different typologies: (i) artificial habitats (tanks, cattle ponds, drinking troughs, troughs, ditches and artificial channels); (ii) springs (springs and resurges with shallow flowing waters); (iii) river, streams and brooks (permanent and temporary waterways); (iv) ephemeral/shallow pools (permanent and temporary small and medium standing water bodies with or without vegetation); (v) marshes, fens and peat bogs (wetlands with peat, mud or mineral soil and different types of vegetation such as flowering plants, sedges, rush or sphagnum mosses and shrubs); (vi) ponds (natural water bodies of small and medium-size also subjected to seasonal changes); (vii) lakes (permanent natural and artificial water bodies).

Biosecurity protocols were strictly followed during field research to avoid spreading emerging infectious diseases (http://www-9.unipv.it/webshi/conserv/monitanf.htm#Norme (accessed on 13 May 2022)).

*2.3. Data Analysis*

Whole observations were first used to outline updated topographic and sub-habitat preferences for the three target species. The altitude and sub-habitat data were collected on-site for each occurrence, and the aspect and slope were derived from the 20 m-precision Digital Elevation Model available from the geo-portal of the Italian Ministry of the Environment (http://www.pcn.minambiente.it (accessed on 7 March 2022)) using the 'Aspect' and 'Slope' algorithms in ArcGIS Pro 2.9 [61]. Then the single-site information was extracted through the 'Extract multi-value to points' tool.

Subsequently, a 10 km² hexagonal tessellation was generated for the study area to show the historical and current pattern of occurrence. We used this data to assess possible changes in species' occurrence, considering the present changes driven by global change [62–65]. Thus, we highlighted "gains" (i.e., where species were not recorded before, thus absent in historical data) and "losses" (i.e., where species no longer occurs after at least two visits).

Considering the evidence of altitudinal shifts that changing climate leads to [62,65–70], we further assessed whether a correlation (Pearson) between the aforementioned changes and altitude occurs, reporting the elevation of the confirmed localities.

We performed two other spatial analyses to support the possible prioritization of territories and conservation strategies. First, we calculated statistically-significant hotspots for each target species using the 'Optimized Hotspot Analysis' tool in ArcGIS Pro 2.9. This approach is based on the Gi* statistics [71] and permits the identification non-random clusters of geographic data by comparing the features of one area to its neighborhood [61,71]. For each geographic element composing the hotspot (or cold-spot, if the cluster has significantly low values), a z-value is also supplied, indicating the number of standard deviations by which the feature differs from the global mean (i.e., the "strength" of the feature within the significant cluster) [61]. As applied in other biology-related stud-

ies [72–74], we applied the False Rate Discovery (FDR) correction to make our results more robust [61,75]. Following the "habitat preferred to the grid" hotspot analysis approach [76,77], we used the spatial data of Corine Land Cover (CLC) (IV level, available at https://land.copernicus.eu/pan-european/corine-land-cover (accessed on 7 March 2022)) and recent occurrence data to run the analyses. Then, we used the tessellation changes' values as weights in Kernel density estimation for each species to predict areas where these gain/stability/loss patterns may have occurred but have not been recorded.

Finally, we assessed the amount of coverage supplied by protected areas for both the kernel densities resulting from the temporal changes in species' occurrence and for the hotspots inferred above. Considering the different goals, we performed the analyses for both Nationally Designated Areas (NDAs, such as Italian National Parks and Regional nature reserves, obtained from http://www.pcn.minambiente.it (accessed on 7 March 2022)) and for Natura 2000 sites (considering exclusively Special Areas of Conservation (SACs), as established by the European Union and downloaded from https://www.eea.europa.eu/data-and-maps/data/natura-13 (accessed on 7 March 2022)).

## 3. Results

During recent field surveys, from 2013 to date, 545 locations were sampled along the study area; target species were present in 264 sites; 77 SACs of the Natura 2000 network also hosted the target species.

When pooling the databases of historical and recent presence records, we observed that Calabrian populations of each target species have peculiar ecological and distributional needs (Figure 1).

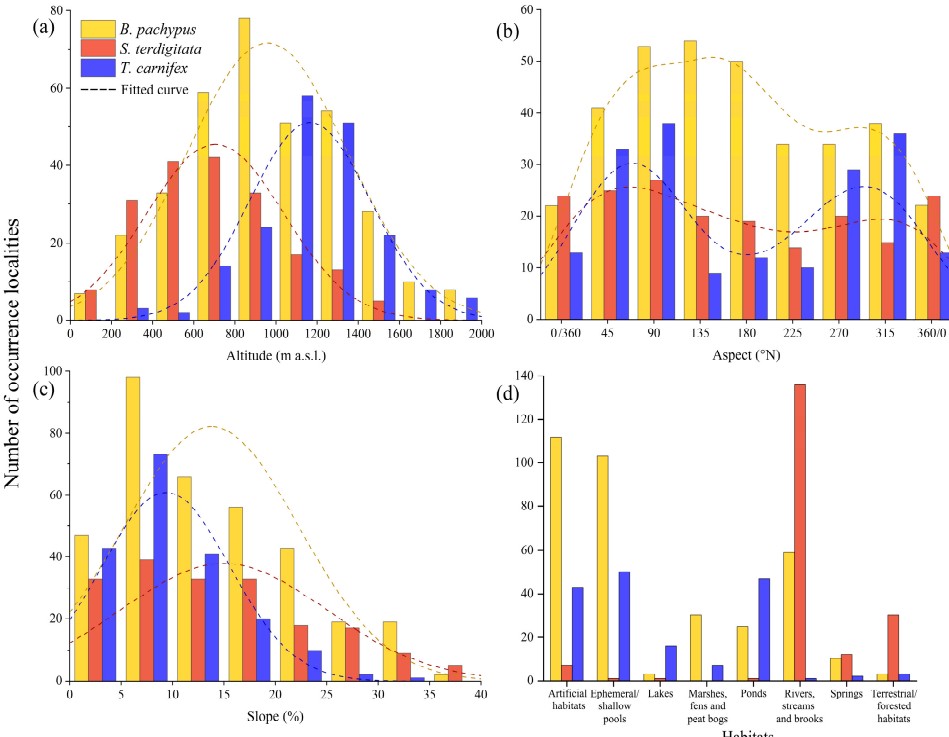

**Figure 1.** Occurrence localities (raw numbers) of *Bombina pachypus*, *Salamandrina terdigitata* and *Triturus carnifex* with their respective distribution for altitude (**a**), aspect (**b**), slope (**c**), and habitat typology (**d**).

For the three target species, we reported the historical, recent, and confirmed sites in each hexagonal cell (Figures 2–4). To support a comprehensive overview of the study area, we also report the Corine Land Cover map (Supplementary Figure S1).

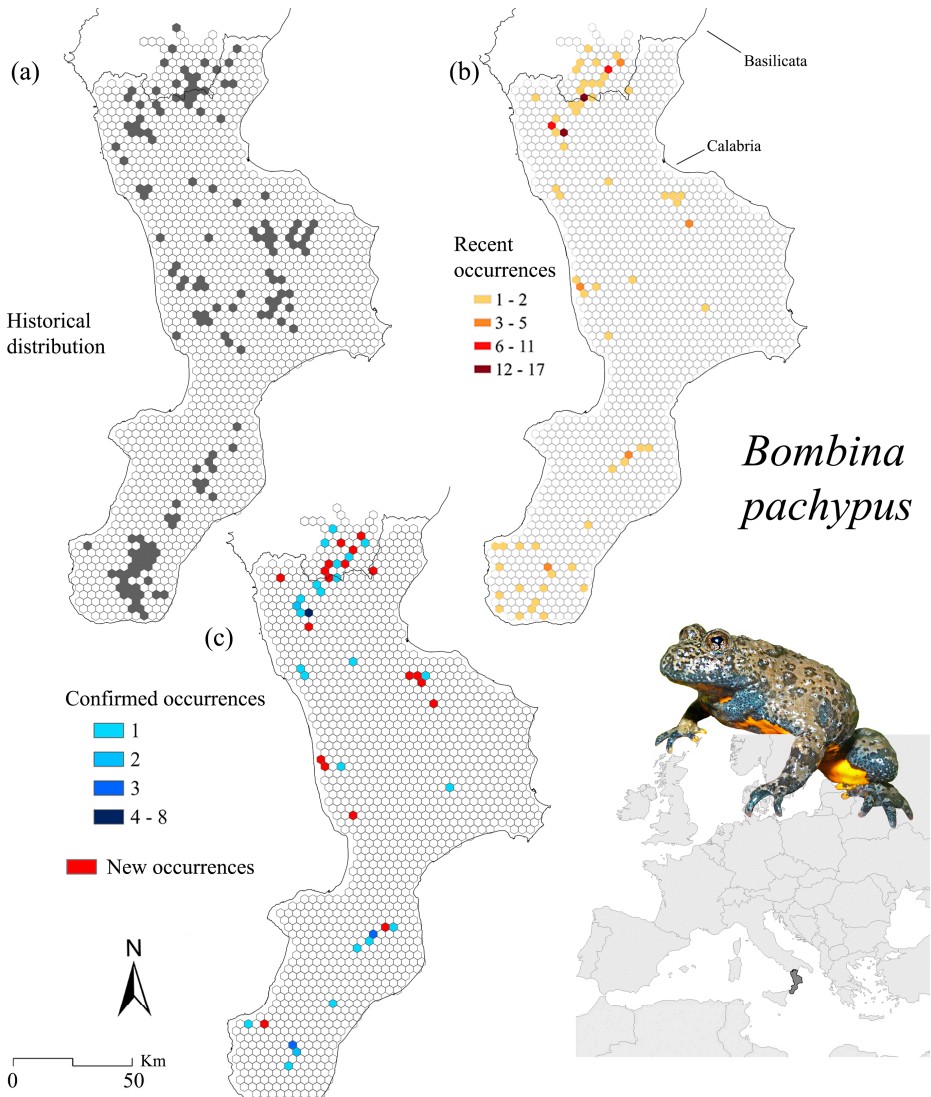

**Figure 2.** *Bombina pachypus* occurrence localities for the historical (**a**) and recent (**b**) distribution, as well as for the confirmed (i.e., known sites where the species still occurs) and new sites (i.e., where the species was found only recently) (**c**).

Comparing altitudinal ranges between sampling periods (1982 to 2006 and 2013–2022) in the study area, the elevation seemed to play a role in changes only for *B. pachypus*, reporting the highest correlation among the three species (Figure 5).

As a general trend, the current hotspot distribution appears to be limited to the main mountain areas of the Calabrian Peninsula, with those inferred over confirmed localities further restricting the ranges (Figure 6a–c). When assessing the composition of these hotspots (in terms of the Corine land cover classes patches they are built upon), we observed a habitat preference generally comparable among the three target species (Figure 6d,e), even though the proportion of single habitats within a certain land cover class may vary (Supplementary Figure S2).

When analyzing the kernel density estimations performed on the historical-recent changes, we found zones where losses and gains overlap, with different magnitudes depending on the specific area and species (Figure 7a–c).

Moreover, we also reported that the percentage of protection offered by the NDAs is generally lower than that provided by Natura 2000 sites, both for kernel density-based losses/gains and for statistically-significant hotspots (Figure 7d,e).

### 3.1. Bombina pachypus

During recent surveys, the Apennine yellow-bellied toad was found in 88 localities, and breeding activity was recorded in 55.7% of the sites. Considering historical and recent data, *B. pachypus* presence records (n = 345) depict a broad elevational range profile from 73 m to 1930 m a.s.l., with most observations occurring between 600 and 1400 m (Figure 1a). The analysis of presence occurrences for *B. pachypus* showed a skewed distribution when considering the aspect, with increasing numbers of records for low values and a peak between N-E and S (Figure 1b). Moreover, the Apennine yellow-bellied toad chooses flat and moderate slopes (Figure 1c). Regarding the habitats used by this species, the most frequent are artificial water bodies and ephemeral or shallow pools (32.5% and 29.8% of the sites, respectively) (Figure 1d).

Figure 2 illustrates the distribution maps of historical, recent, and confirmed presences. The number of 10 km² cells in which historical sites fall is 144, whereas recent occurrences cover a total of 62 cells (Figure 2a,b); we also found 66 new occurrence records with respect to the previously known ones. The sites where the species still occurs total 39 (falling into 27 cells), accounting for 15.2% of historical point records in localities distributed across the study area (Figure 2c).

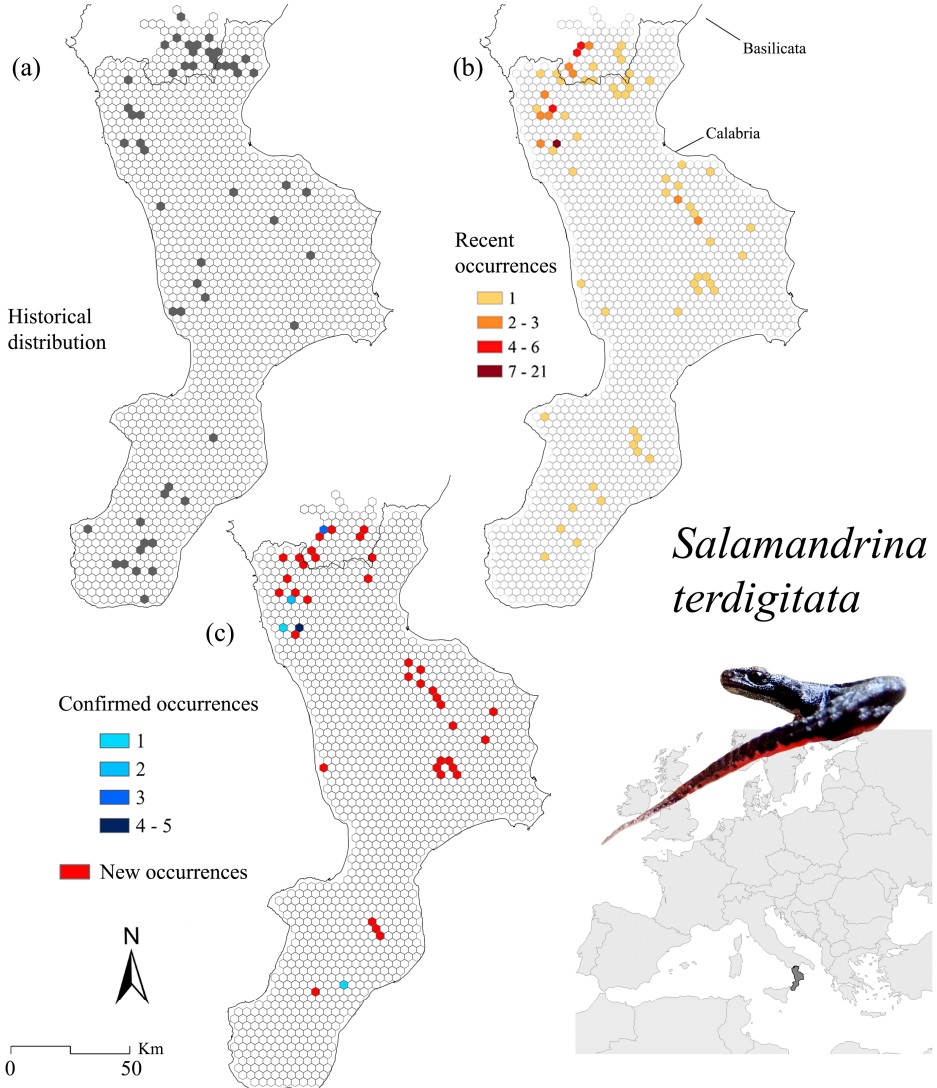

**Figure 3.** *Salamandrina terdigitata* occurrence localities for the historical (**a**) and recent (**b**) distribution, as well as for the confirmed (i.e., known sites where the species still occurs) and new sites (i.e., where the species was found only recently) (**c**).

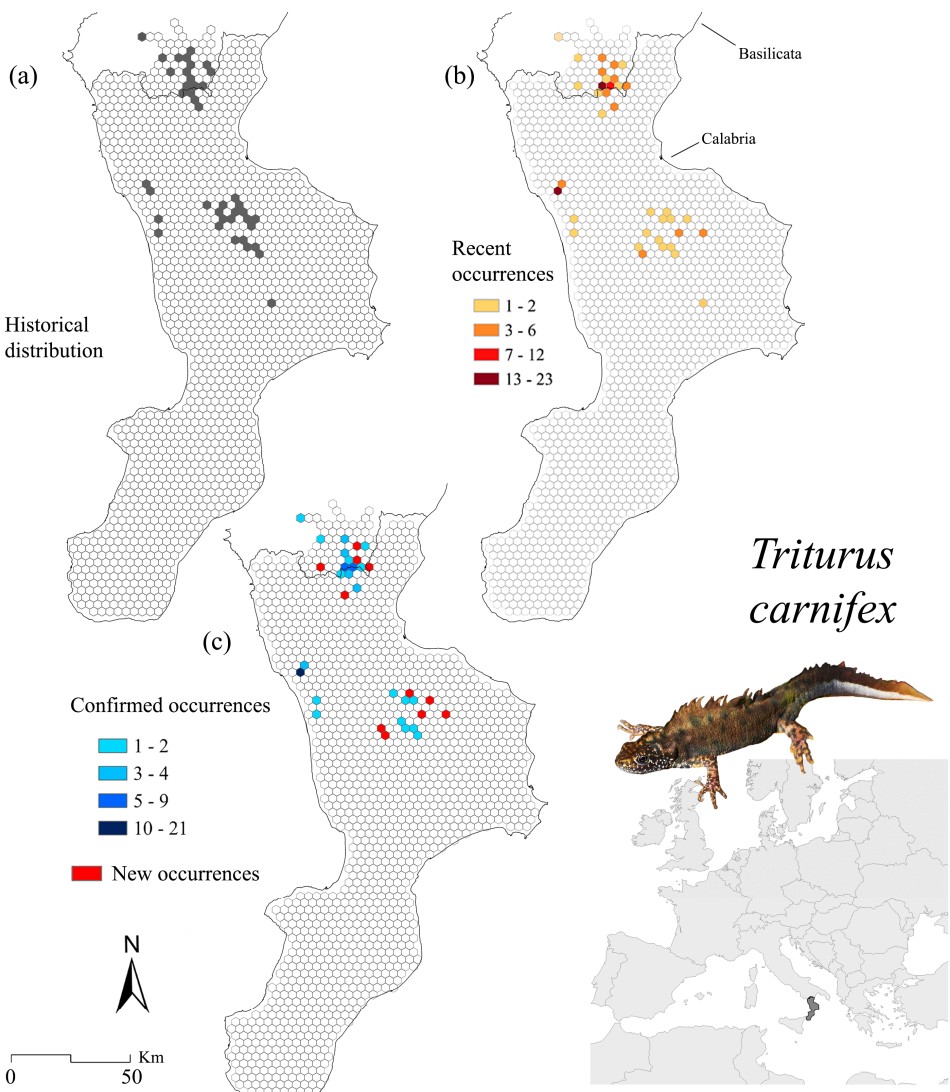

**Figure 4.** *Triturus carnifex* occurrence localities for the historical (**a**) and recent (**b**) distribution, as well as for the confirmed (i.e., sites where the species still occurs) and new sites (i.e., sites where the species was found only recently) (**c**).

When measuring the possible relationship between changes in presence sites and elevation, we detected a moderate correlation ($R^2 = 0.46$, $p = 3.9 \times 10^{-24}$) (Figure 5a), finding a higher number of confirmed localities in altitude preferred by the species (Figure 5d).

We obtained a similar spatial arrangement of statistically-significant clusters from the hotspot analysis for both recent records and confirmed sites (Figure 6a). These are mainly concentrated in the central and southwest portion of the Pollino massif and the south core of the Aspromonte region; sub-cluster areas with less statistically-significant value have been identified along the Catena Costiera, the north-east sector of the Sila plateau and the Serre massif, and areas in the southernmost part (province of Reggio Calabria), both on the Tyrrhenian and Ionian sides (Figure 6a). Through the analysis of the CLC categories which compose the identified hotspots, it is evident that *B. pachypus* equally occurs in "Land principally occupied by agriculture with significant areas of natural vegetation" (LC code 243) and "Transitional woodland and scrub" (LC code 324) (Figure 6d,e). As a general trend, about 27% is represented by shrubs and/or herbaceous vegetation, followed by forests (26%) and agricultural areas (22%, including pastures and heterogeneous patches).

Kernel density analysis revealed areas with possible species presence changes in all the study region's relevant zones. Major gains were predicted on the western side of the Pollino

National Park, along with some minor ones in the southern portion of the Catena Costiera, northern Sila (Sila Greca), and further south in the Aspromonte massif (Figure 7a). Major losses were concurrently inferred in the same mountain regions, with a higher magnitude and often overlapping gains (Figure 7a). We found a similar coverage percentage for kernel-based gains and losses for both NDAs and Natura 2000 sites (Figure 7d). When considering the hotspot analysis results, the proportion of areas protected under the Habitats Directive was higher than NDAs (Figure 7e). Likewise, our results showed comparable coverage if considering the inferences calculated over confirmed occurrence localities (Supplementary Figure S3).

### 3.2. Salamandrina terdigitata

The Southern spectacled salamander was observed at 102 localities, and breeding activity was recorded in 50% of the sites. Considering the whole dataset (n = 189), the elevational pattern shows that *S. terdigitata* is distributed over a wide altitudinal range, spanning from 80 to 1550 m and with a peak between 200 and 1000 m a.s.l. (Figure 1a). Topographic analysis reveals a bimodal distribution, with a slight preference for NE–E and W–NW exposure (Figure 1b). Regarding slope, occurrences cover a range not exceeding 30%, with most ranging from 0–20% (Figure 1c). The species was mainly found in rivers, streams and brooks, both permanent and temporary (71.9%), and lower percentages in terrestrial or forest environments (15.8%) (Figure 1d).

Historical, recent, and confirmed distribution ranges are summarised in Figure 3 (historical sites' cells = 52, recent sites' cells = 65; Figure 3a,b, respectively). We recorded a total of 79 new records, whereas we have confirmed 23 presence sites which account for the 26.4% of known sites, occurring within 5 cells (Figure 3c). We found no significant correlation ($R^2 = 0.02$) between changes in occurrence localities and elevation (Figure 5b); almost all confirmed sites ranged in the 400–800 m a.s.l. (Figure 5e).

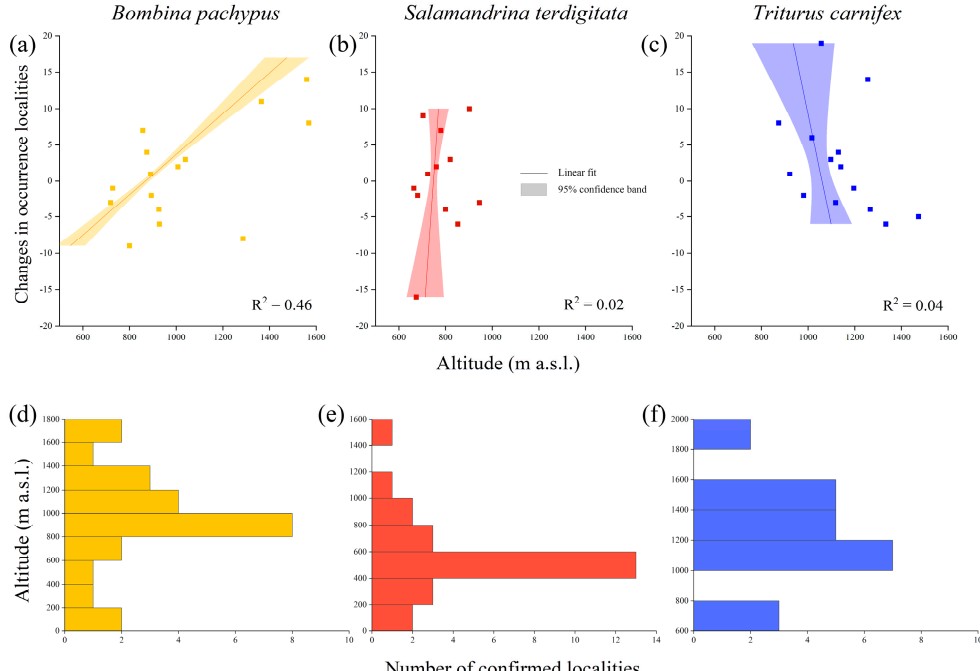

**Figure 5.** Correlation between changes in occurrence localities (historical-recent records, tessellation) for *Bombina pachypus* (**a**), *Salamandrina terdigitata* (**b**) and *Triturus carnifex* (**c**). From the same data, the altitude of confirmed records (i.e., occurrence localities where the species still occurs from historical times, since ~1980) is reported for *B. pachypus* (**d**), *S. terdigitata* (**e**) and *T. carnifex* (**f**).

Hotspot analysis reports spread and discontinuous patches, with neat high clustering missing. In both recent- and confirmed-based hotspots, the Orsomarso mountains and the northern part of the Catena Costiera stand out; in contrast, the Sila, Serre and Aspromonte

hotspots have slightly lower z-values (Figure 6b). The corresponding land cover percentage indicates that *S. terdigitata* equally inhabits heterogeneous agricultural areas (LC code 211, 242, and 243, for a total of 26.9%) as well as forest (26.3%) and transitional woodland/shrub areas (mainly LC code 324, 26.6%) (Figure 6d,e).

Kernel density estimations performed over putative changes revealed the highest losses in the northeast of the Pollino massif. At the same time, gains were inferred in the Tyrrhenian ridge of the massif, with some degree of overlap between each other. Minor losses and gains were also reported in Sila Greca, southern Catena Costiera, and the southeastern tip of the Aspromonte (Figure 7b). The protection offered by NDAs and Natura 2000 sites varies greatly when considering the kernel-based changes, with the former covering less than a half compared to the latter (Figure 7d). Similarly, the Natura 2000 sites cover a higher percentage of hotspots (Figure 7e), a trend also found for the confirmed sites-based hotspots (Supplementary Figure S3).

### 3.3. Triturus carnifex

During the recent surveys, the presence of *T. carnifex* was ascertained in 74 sites (37 cells), with 56.7% representing breeding habitats. Based on a total of 168 occurrence points, *T. carnifex* occurs over a broad altitudinal range, with the highest distribution of up to 1848 m; observations peak at elevations between 1000–1400 m a.s.l., and no observations occur below 200 m (Figure 1a). Similarly to *S. terdigitata*, *T. carnifex* shows a bimodal distribution for aspect, preferring NE-SE and W-NW exposures (Figure 1b) and gentle slopes (Figure 1c). It has been observed in a wide variety of habitats, and ephemeral or shallow pools (29.8%), ponds (28%), and artificial aquatic habitats (25.6%) represent the most frequent (Figure 1d).

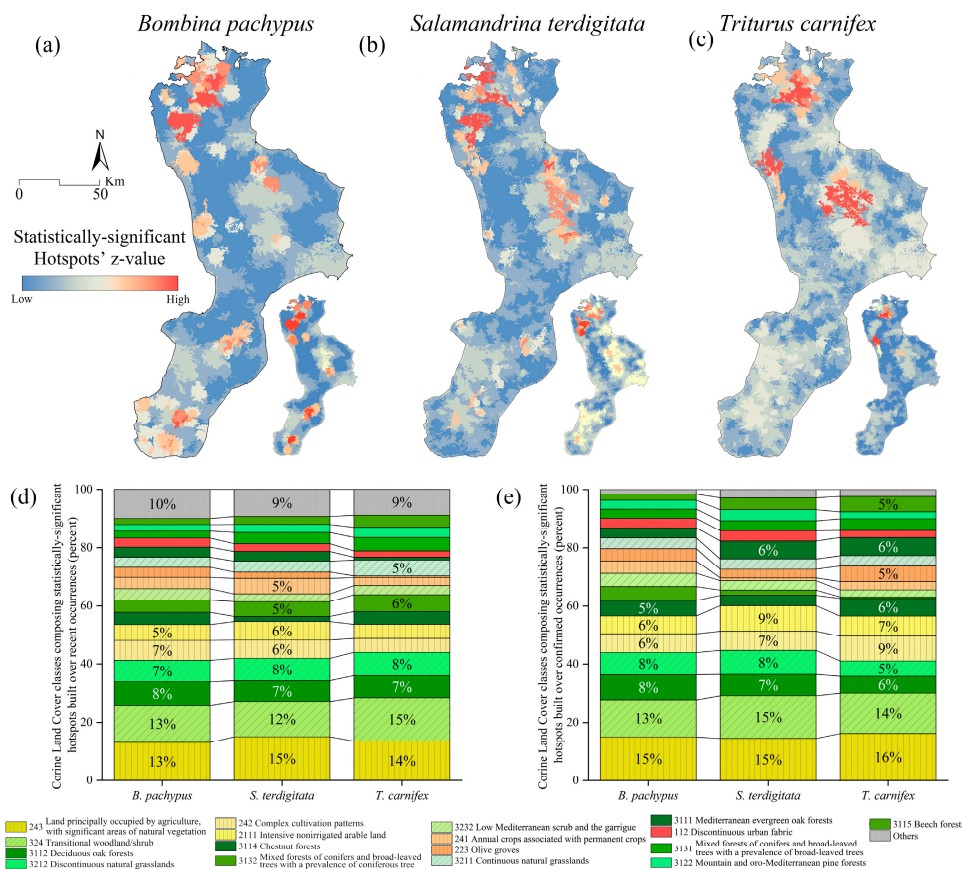

**Figure 6.** Statistically-significant hotspots built over the recent (large map) and confirmed (small map) localities and Corine Land Cover (IV level) patches for *Bombina pachypus* (**a**), *Salamandrina terdigitata* (**b**) and *Triturus carnifex* (**c**) and their composition (higher classes, in terms of z-values) for recent (**d**) and confirmed (**e**) data.

Historical, recent, and confirmed distributions over the three known cores in the southern part of *T. carnifex* range are given in Figure 4. Historical data (n = 94) fall in 41 cells, while current records cover 36 cells (Figure 4a,b, respectively). We recorded 34 new examples of data on the presence of this newt species. Our investigations confirmed occurrence in 57 sites (falling into 24 cells), corresponding to 60.6% of the previously known localities (Figure 4c). Comparing historical elevation data to recent ones, no significant changes were detected for *T. carnifex* ($R^2$ = 0.04) (Figure 5c); the altitudinal intervals of confirmed presence sites also showed a wide variation (Figure 5f).

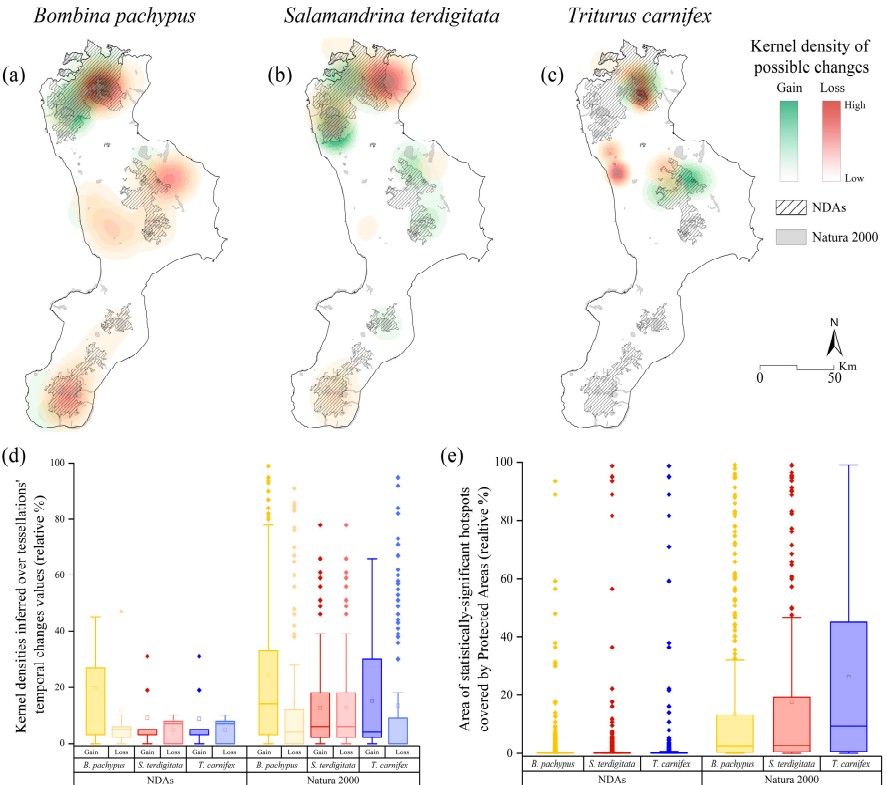

**Figure 7.** Kernel density estimation of possible areas of gains and losses inferred on historical/recent records (tessellation) for *Bombina pachypus* (**a**), *Salamandrina terdigitata* (**b**) and *Triturus carnifex* (**c**). Protected areas coverage for the three target species in terms of kernel density estimates (**d**) and hotspots (**e**), calculated for both Nationally Designated Areas (e.g., National Parks, Reserves) and Natura 2000 sites.

Three main cluster areas emerge from the hotspot analysis: the central-eastern portion of the Pollino massif, the Catena Costiera (these two are also involved in the confirmed occurrences-based hotspots) and the Sila plateau (Figure 6c). The corresponding land cover classes from spatial analysis describe a preference for the transitional areas (31.6%, mainly the "Transitional woodland/shrub", LC code 324) followed by forests (30.6%, primarily the "Forests dominated by deciduous oaks", code 3112) and heterogeneous agricultural areas (21.3%, mainly in the natural or semi-natural areas, code 243) (Figure 6d). Considering hotspots built over confirmed occurrences, we detected a slight increase in areas covered by both heterogeneous agricultural areas (LC codes 2111, 241, and 243) and forests (3112) (Figure 6e).

The kernel density estimations for gains and losses involve the same hotspot areas, but specific trends for each zone were detected (Figure 7c). Pollino and Sila reported overlapping gains and losses, while Catena Costiera revealed only losses.

Finally, the protected areas' coverage analysis displays a similar trend to *S. terdigitata* in terms of the difference between NDAs and Natura 2000 percentages for kernel-based changes (Figure 7d), as well as for the hotspots (Figure 7e). The Natura 2000 network supplies higher protection, as also observed in the case of confirmed occurrences (Supplementary Figure S3).

## 4. Discussion

Our findings provide up-to-date distribution, baseline information on the conservation status, and spatially explicit information for *B. pachypus*, *S. terdigitata*, and *T. carnifex*. These data may help support conservation strategies in southern Italy, a well-recognized biodiversity hotspot [2,3,9,23]. The recently published research on these three relevant species of Italian fauna in the south part of their range has focused mainly on genetic analyses, phylo-geography, and the occurrence of the pathogenic fungus *Batrachochytrium dendrobatidis* [3,23,39,46,78]. Therefore, previous works focusing on distribution, ecology, and assessment of the conservation status for the three target species in this crucial area are rather dated [35,56–58,79] or not extensive enough to portray a current evaluation of the status, particularly for *B. pachypus*.

### 4.1. Bombina pachypus

Severe concerns weigh on the conservation status of the Apennine yellow-bellied toad, considering the extensive population declines estimated from 50–80% [80]. Evidence indicates several drivers: the loss or degradation of breeding habitats, climate change, and an uncertain role of chytridiomycosis [17,22,23,78,81,82].

According to the data collected locally starting from the early 1980s, the distribution ante 2006 of the Apennine yellow-bellied toad in the study area was relatively uniform across the Pollino massif to the Aspromonte chain with 257 known sites. As reported by Andreone et al. [80] and Barbieri et al. [81], the status of *B. pachypus* populations in the southernmost portion (the Calabria region) constituted an exception if compared to the dramatic demographic decline of populations in the rest of its range. Currently, the question is whether the populations from this area still have a favourable situation or if it is reasonable to expect a contraction in the number of presence sites, demographic reduction and population loss, as previously indicated by several authors [23,83,84]. Our results also suggest that recent population decline and local extinctions have occurred in the southernmost part of the Apennine yellow-bellied toad range with different declining trends along the study area.

Our thorough field surveys in the area protected by the Pollino National Park depict a not alarming scenario compared to Canestrelli et al. [83]. Indeed, even though *B. pachypus* distribution appears less spread than in the past [35,57]; the species is nowadays mainly localized on the Tyrrhenian side, in the Orsomarso massif and on the eastern side in Basilicata. Also, when considering the extreme tip of the study area (the Aspromonte region and its neighboring zones), our results are in line with the overall favourable status outlined by other fine-scale field surveys of *B. pachypus* populations assessing their consistency and genetic variability [23,85]. These trends are supported by historical data, recent records, and spatial statistics. Our statistically-significant hotspots mainly fall within the Pollino and Aspromonte National Parks and Serre Regional Park, even though the overall protection is low, considering the broad "cold" patches surrounding the "hottest" ones. Considering the possible changes over 40 years, the areas with the highest kernel density-based gains and losses always fall within both NDAs and Natura 2000 network.

The most alarming finding of this research is the *B. pachypus'* populations missing from several historic sites, especially in the broad area covered by the Sila National Park and the Sila Biosphere Reserve of the World Network of UNESCO sites of excellence. In these areas, some recent investigations (2017–2022, mainly technical reports [86–89]) are bringing to light this decline; the recent occurrence of *B. pachypus* is confirmed only in a few sites in Sila Greca and Sila Piccola. The increase in cattle grazing, pollution, human recreational activities in the habitats used by the species, and the introduction of exotic species could pose a severe threat to the amphibian in this area.

Moreover, the elevational distribution of *B. pachypus* may have shifted compared to the historical presence records (dating up to 40 years ago). In the '90s, the upper elevation limit of *B. pachypus* was reported at 1930 m a.s.l. in a network of pools on the Piana di Pollino/Mandre del Tarantino [35,58]. This occurrence was not confirmed during recent

surveys, and the highest altitude newly recorded is ~1700 m a.s.l. Therefore, we strongly recommend directing specific surveys in this protected area, including the Catena Costiera. In parallel, we suggest planning a disease screening on sampled population mainly in the Sila plateau, but also at a regional scale, as reported by other authors [25,83]. Pathogen outbreaks have been present in Calabria since the late 1970s [78] and recently samples of *B. pachypus* tested positive for *Batrachochytrium dendrobatidis* in the Catena Costiera [23], Pollino [25] and Sila National Park (unpublished personal data).

When dealing with general ecological needs and corresponding possible conservation strategies, our study confirmed that *B. pachypus* is characterized by a relatively wide ecological niche. The Calabrian populations prefer wooded environments and agricultural land, pasture and scattered patches of natural/semi-natural areas in the hilly-mountain plain and mountain sides with the sunniest exposure. Our analysis also showed that, as much as the territorial availability of the aspect encompasses the whole exposure range (result not shown), the species distinctly prefers the NE–E and W–NW. The Apennine yellow-bellied toad occurs in many types of wetlands but with a marked preference for non-shaded temporary pools and artificial water bodies associated with human activities (e.g., cattle and irrigation ponds, tanks, drinking troughs). This information agrees with the known habitat preferences for the species, especially in the southern Apennines [17,19,42]. About 20% of the non-confirmed historical sites recently visited are artificial water breeding sites abandoned or destroyed in a rural context. Therefore, to halt the negative trend and preserve this endangered amphibian, solid conservation strategies and management actions at the local scale must comprise the restoration or creation of artificial and semi-natural water habitats. These measures should be coupled with ex situ programs (i.e., captive-breeding, reintroductions and translocations), following the best practices already accomplished in several projects [90–93].

In this context, developing a national action plan for establishing priorities in the Apennine yellow-bellied toad conservation is mandatory. An action plan could help build up an experts' network, improve coordination of conservation efforts across the species' range, and provide evidence-based recommendations sharing different knowledge. Notably, the populations of *B. pachypus* in the southernmost portion of the range could be of crucial conservation value for two main reasons: (1) Calabria hosts populations with the highest levels of genetic diversity than in the rest of the Peninsula [3,23], and (2) the status of this species seems to be less dramatic than elsewhere in its range, even in future predictions [37].

### 4.2. Salamadrina terdigitata

*Salamandrina terdigitata* in southern Italy, and *S. perspicillata* from central to the northern Apennines, represent a unique case of a species of an endemic amphibian genus of Italy [40].

Across the study area, the historical distribution of *S. terdigitata* appeared to be non-uniform and discontinuous and represented by 87 occurrence points. Thanks to recent surveys, we found many new sites, especially on the Tyrrhenian side of the study area and in the eastern portion of the Sila plateau and of the Serre massif. Therefore, this research contributes to better defining the distribution of *S. terdigitata* in the southern Apennines, providing helpful information for a future assessment of its conservation status, currently not updated [43]. Comparing historical and current data, the distribution of *S. terdigitata* in Calabria appears stable; however, there is no information on the size of the populations.

Our findings partially fill the knowledge gap in the ecological traits of this species. In fact, the available literature mostly refers to *S. perspicillata* [94,95]. In the study area, the species covers a wide altitudinal range; we reported two new breeding sites, both on the Ionian side, representing the lowest record (Valle del Trionto, Sila Greca, 80 m a.s.l.) and the highest (Vale del Torrente Soleo, Sila Piccola, 1550 m a.s.l.) for *S. terdigitata*.

The topographic analysis revealed that locations with both northeastern and north-western exposure, and flat and moderately sloped inclinations, offer optimal conditions for *S. terdigitata*. This confirms that complex topography is essential in determining the availability of refugia and suitable microclimate necessary for this species [95]. Our obser-

vations indicate that *S. terdigitata* occurs within mixed deciduous broad-leaved forests but also in semi-natural and agricultural areas characterized by the presence of running waters. Breeding sites were mostly found in pools near slow-running streams (39%); the southern spectacled salamander was sporadically found in other environmental categories, such as artificial water environments, as previously reported [35,42].

Improving knowledge of the distribution and consistency of populations, especially in the northeastern part of the Pollino massif and on the Ionian side, is recommended. The Calabrian Peninsula shows sharp climatic contrast due to the geographic position and orography. Specifically, for the two coastal sides, the Tyrrhenian one is influenced by western air currents, which cause milder temperatures and intense precipitations; on the contrary, the Ionian side is exposed to the warm African winds leading to short and heavy rainfall [96]. Since this species requires stable climatic conditions [6] and future projections suggest that climatic suitability will be heavily reduced [24], our results indicate that further comprehensive data gathering is needed in the areas on the Ionian side. Here, changes in climatic suitability could determine the disappearance of temporary aquatic reproductive sites.

Additionally, more details on the reproductive biology and ecological requirements of the southern spectacled salamander are needed, considering that these still remain vague if compared those of *S. perspicillata*.

To maintain a favorable conservation status, sustainable management of forested habitats characterizing the landscape of the study area is fundamental. Local stakeholders, especially protected areas' managers, should pursue habitat conservation actions such as keeping abundant deciduous leaf litter, stumps, and logs and avoiding forest clear-cutting and coppicing. Water environments should not be impacted by fish introductions, drainage, and pollution.

### 4.3. Triturus carnifex

In the southern borders of the range, the distribution of *T. carnifex* is clustered and restricted to the northern Calabria mountains, having not been found south of the Sila plateau. The comparison of our recent data with those of the historical database and other published studies [35,56,79,97] basically confirmed the species' distribution at the southern limit of its range. We found the Italian crested newt populations to mainly occur on the mountainous reliefs, from 600 m to up 1700 m in Sila, and up to 1845 m on the Pollino massif [56,58].

According to previous studies focusing on our same area, *T. carnifex* was associated with a wide variety of habitats from deciduous, mixed or coniferous forests to xeric environments but also in cultivated and semi-natural areas [28,36]. We found a quite similar occurrence between northeastern and northwestern facing slopes and a preference for slight to moderate slopes. *T. carnifex* reproduces in permanent and temporary wetlands but it is also frequently found in artificial water bodies such as tanks and drinking troughs [42,58].

Our spatial analysis highlighted a low coverage percentage of the populations of *S. terdigitata* and *T. carnifex* within protected areas and showed a loss of sites within the areas covered by NDAs. The gains emerging after the recent investigations highlighted more extensive hotspot coverage within the Natura 2000 network, conceivably because the sites are spread across the entire study area.

In Italy, *T. carnifex* shows a decreasing trend [45]; one primary pressure on the species is the alteration or loss of reproductive habitats due to agricultural intensification and pollution [18,97]. In northern Italy, Falaschi et al. [16] recently reported a dramatic decline in the abundance of this newt related to invasive fish and crayfish. Similarly, the Catena Costiera lakes and ponds were impacted by the recent introduction of invasive alien fish (*Gambusia* sp. and *Cyprinus carpio*, pers. obs.), posing severe worry about the relevant biogeographic and conservation value of amphibian communities occurring there. As stated above regarding *B. pachypus*, the local management of artificial habitats in the rural Calabrian landscape should be a priority for the local managers.

## 5. Conclusions

Since amphibians are undergoing a rapid decline with high rates of species loss, one of the most crucial conservation activities is monitoring populations' distribution, abundance, and dynamics, to better integrate conservation strategies and management approaches. One issue in understanding changes in conservation status and trends over time arises from the lack of updated studies at both local and regional levels.

Here, we provided an up-to-date knowledge framework on the distribution, conservation, and ecological needs of three amphibians inhabiting the southern Italian Peninsula; among these, *B. pachypus* represents a conservation emergency because it is one of the vertebrates at greatest risk of extinction in Italy. Thanks to the availability of a valuable and robust set of historical data, we were able to make comparisons and spatial and ecological inferences on population trends over time of the investigated species.

The overall findings of this work outline that the patterns of species occurrence in the southernmost part of Italy did not dramatically change in recent decades. Although we detected a marked difference in the number of sites of presence (historical and new sites) in some sub-areas of the region, the three investigated species could be least affected by the severe population declines reported in other parts of their ranges.

Our study indicates that Calabria, notably the Pollino National Park, still retains suitable habitats for three amphibian species in the southern Apennines. In particular, one of the most valuable outcomes we found regards *B. pachypus*. In fact, the whole results obtained for this species pinpoint some priority areas in which to concentrate greater survey effort and set conservation plans, such as the restoration of the artificial aquatic breeding habitats, ex situ breeding, and translocation programs.

**Supplementary Materials:** The following supporting information can be downloaded at: https://www.mdpi.com/article/10.3390/land11081292/s1, Figure S1: Corine Land Cover map of the study area (III level); Figure S2: Proportion of the aquatic habitats (both natural and artificial) recorded in the field occurring within the Corine Land Cover patches used for the present research; Figure S3: Protected areas coverage for the three target species in terms of confirmed localities-based kernel density estimates (a) and hotspots (b), calculated for both Nationally Designated Areas (e.g., National Parks, Reserves) and Natura 2000 sites (SACs); Table S1: Dataset used for the presented analyses containing the occurrence localities (decimal degrees, WGS84) for *Bombina pachypus*, *Salamandrina terdigitata* and *Triturus carnifex*.

**Author Contributions:** Conceptualization, I.B. and M.I.; methodology and formal analysis, I.B. and M.I.; data collection I.B., V.C., S.T., V.M., S.P.; data curation, I.B.; writing—original draft preparation, I.B. and M.I.; writing—review and editing, I.B., M.I., S.T. and M.B.; supervision, S.T. and M.B. All authors have read and agreed to the published version of the manuscript.

**Funding:** This research was funded by Ministero dell'Istruzione, dell'Università e della Ricerca-A.I.M. Project PON R & I 2014–2020, grant number 1870582 and in part by Research Grant number 103/2021 funded by Regione Calabria, FESR-POR Calabra 2014–20120-Azione 6.5.A.1-Sub Azione 1.

**Institutional Review Board Statement:** Field surveys and sampling procedures were approved by the Italian Ministry of Ecological Transition (permit number: PNM-2018-0012568 and PNM-2022-0008263).

**Informed Consent Statement:** Not applicable.

**Data Availability Statement:** All data used for the analyses are available in the Supplementary Materials as reported in the main text.

**Acknowledgments:** We are grateful to the students from the University of Calabria, Quirino Valvano, Eleonora Bozzo and Greta Spinelli, who assisted in the field research. Emilio Sperone, Antonio Mancuso, Marco Bologna, Edoardo Razzetti, Fabrizio Oneto, Gigi Poerio, Francesco Leonetti and Gianni Giglio kindly provided useful data.

**Conflicts of Interest:** The authors declare no conflict of interest.

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
