# Peer review of "Updating Distribution, Ecology, and Hotspots for Three Amphibian Species to Set Conservation Priorities in a European Glacial Refugium"

_land, doi:10.3390/land11081292_

Round 1

Reviewer 1 Report

In figures 2-4 an administrative division layer or coastline should be added to help locate the hexagonal grids.

Author Response

Dear Reviewer, 

thank you for your comments. Here we report the corresponding response:

In figures 2-4 an administrative division layer or coastline should be added to help locate the hexagonal grids.

- Corrected. We added the required information in the figures.

Reviewer 2 Report

This study introduced the distribution of three amphibian species in Southern Italy, analyzed their habitat preferences, and performed a gap analysis for conservation effectiveness. The literature review of this study is sufficient, and the data are very solid. I believe the results are convincing. I have a few comments listed below:

1.     Line 210. The land cover data “Corine Land Cover (CLC)” were used to support the conservation priority areas. More information is needed for this dataset. How to compare CLC with eight wetland types in Figure 1d?

2.     Figure 1 shows the probability density of the three species along the gradient of four variables. I suggest to use points rather than bars. Points show the original data, whereas bars represent grouped data. A random number (Y-axis value) at the range of 1-100 can be given to each point in order to spread out the points. To make the density curve clearer, please use solid thinker curves to replace dashed thin curves. This is not a mandatory request. The panel d can remain unchanged.

3.     Figures 2-4 clearly show the historical and current occurrences of the three species, as well as the comparisons between the two stages. I also expect a map of landcover for the study area, showing the actual situation of the area to the readers who are not familiar with the local conditions. This map can be added as a supplementary figure.

4.     The Panels a-c in Figure 5 look weird, as the sums of residuals are over zero, not equal to zero, i.e. the fitted lines locate too low in the figures.

5.     Figure 6 shows the Corine Land Cover (CLC) classes. Please use ecological meaningful names for the classes.

6.     Lines 332-333. “Exposure analysis reveals a bimodal distribution, with a slight preference for NE–E and W–NW, with the highest values occurring at 90° and 360° (Figure 1b).”. Do you think such bimodal distribution means anything? The slope aspect is associated with sun radiation, which might meaningful to these amphibian species. Anyway, what is the availability of the aspect? I guess there are large areas at the aspect of NE–E and W–NW, so that is why the animals occur more at such aspect. What is the odds of the habitat use? On the other hand, what is the difference between 0 and 360? They are identical!!! I always feel using degree to quantify direction is a mistake. You may use a north index, transferring 0 and 360 degrees to 0, transferring 180 degree to 1, and changing 90 and 270 degrees to 0.5.

Reviewer 3 Report

I find this publication well written and interesting, thus worth to publish. I have the following comments and suggestions:

I think it would be gain more interest if you would change the title to „…three amhibian species of European Community interest…” as it is better describes why you chose these species.

According to well established knowledge on Bombina phylogeograpghy Bombina pachypus is not a valid species (for example see the recent article by Dufresnes et al. 2020, doi:10.1111/jbi.14018). Please refer for this species as B. variegata and may write about B. pachypus as a former species with its own IUCN and Habitats Directive category.

Line 125: change to protected areas

Line 195: It is very hard to clarify that a species is no longer present in an area and 10 km2 hexagons are very large. It may be possible that you missed some of the unknown localities. It would be informative to write one or a few sentences that you define ’loss’ if you didn’t find the certain species after two visits to former localities. Did you tried to find new localities in hexagons where there was know localities?

Figure 6. It would be much better to write the names of the land cover categories and not just their codes because it is hard to interpret for the reader. I think that there are enough space for that and I strictly recommend this. Maybe it is a solution to put the CLC classes legend under the d) and e) figures.

Line 548: change to northwestern

Author Response

Dear Reviewer,

thank you for your comments. Here we report our responses: 

I find this publication well written and interesting, thus worth to publish. I have the following comments and suggestions:

 - We thank the Reviewer for the work on our manuscript and for the very useful comments and suggestions.

I think it would be gain more interest if you would change the title to „…three amhibian species of European Community interest…” as it is better describes why you chose these species.

- Of course, the title would be more informative in the way you suggest, but it is already packed with information and three-lines long! Thus, we would avoid adding further words to it.

According to well established knowledge on Bombina phylogeograpghy Bombina pachypus is not a valid species (for example see the recent article by Dufresnes et al. 2020, doi:10.1111/jbi.14018). Please refer for this species as B. variegata and may write about B. pachypus as a former species with its own IUCN and Habitats Directive category.

- Since there is an ongoing debate about the taxonomic status of this species, and even recent papers dealing with it use B. pachypus to identify this species, we prefer (also considering the conservation and local distribution purposes of our research) to declare this issue, leaving to the reader the possibility of deepening the knowledge about taxonomy with the published literature, and keeping the pachypus species name to be consistent with the national literature. In fact, we added a whole paragraph to discuss this issue, L 97 – 102.

Line 125: change to protected areas

- Corrected.

Line 195: It is very hard to clarify that a species is no longer present in an area and 10 km2 hexagons are very large. It may be possible that you missed some of the unknown localities. It would be informative to write one or a few sentences that you define ’loss’ if you didn’t find the certain species after two visits to former localities. Did you tried to find new localities in hexagons where there was know localities?

- We already declared the “two visits” method in L 180-181; indeed, we added a further clarification in L 209-210.

Figure 6. It would be much better to write the names of the land cover categories and not just their codes because it is hard to interpret for the reader. I think that there are enough space for that and I strictly recommend this. Maybe it is a solution to put the CLC classes legend under the d) and e) figures.

 - Corrected. We added the required information in Figure 6 as suggested also by Reviewer 2.

Line 548: change to northwestern

- Thanks for noting this typo, we have corrected it.

Reviewer 4 Report

The manuscript presents interesting approach to amphibian ecology and conservation. The methods are adequate (although they should be described in more details, see comments) and the results are clearly presented. The discussion suffers from some purely speculative statements, and the supplementary table is incomplete, with reversed coordinates of the species records. Additional comments:

Lines 109-111: Written in this way, the passage suggests that T. carnifex only occurs in Italy, which is not true - provide more precise distribution.

Lines 157-158: The sources are not reported in Table S1 - it contains only species, database type and X/Y coordinates. The coordinates themselves are reversed, pointing to the Red Sea instead of Southern Italy.

Lines 163-166: Funnel traps are standard method for detecting aquatic species in general, and newts in particular - but were not used for these field studies... Why?

Lines 218-220: Considering your results, it would be interesting to compare these in terms of coverage per area. If the SACs have much larger area than the NDAs, than that could explain why they contain more localities (i.e., they don't necessarily cover more localities per area).

Lines 251-253: Were funnel traps used in the "old" sampling period? If so, comparing these results with only visual searches of the recent surveys might not be very reliable.

Lines 284-285, 328-329 and 362-363: Were all sites visited during the breeding season (unlikely, considering that visits were conducted until late summer)? If not, the reported "breeding activity" is not reliable. Provide more details.

Lines 447-451: This is an interesting topic for future research, but at this point I think it is pure speculation and should be removed.

Lines 455-458: Actually it is the snow-line that matters for the rate of warming, not a specific altitude. Again, if you can't provide a reference to back your claim, I suggest you remove it.

Author Response

Dear Reviewer,

we appreciated your comments and suggestions, to which we responded as reported below: 

Comments and Suggestions for Authors

The manuscript presents interesting approach to amphibian ecology and conservation. The methods are adequate (although they should be described in more details, see comments) and the results are clearly presented. The discussion suffers from some purely speculative statements, and the supplementary table is incomplete, with reversed coordinates of the species records.

- We thank the reviewer for the positive feedback and the useful comments on the manuscript. Below, we report our response to your comments, which were all addressed and solved.

Additional comments:

Lines 109-111: Written in this way, the passage suggests that T. carnifex only occurs in Italy, which is not true - provide more precise distribution.

- Thanks for this suggestion. We have specified the Italian distribution in L 121-123.

Lines 157-158: The sources are not reported in Table S1 - it contains only species, database type and X/Y coordinates. The coordinates themselves are reversed, pointing to the Red Sea instead of Southern Italy.

- About this issue, we noticed our coordinates properly falling within the study area, but found that if copied and pasted on, say, google maps, they fall into the Red Sea…but just because Google maps accepts the Y,X format. If opening the Supplementary table with a GIS and selecting our X column as longitude and Y as latitude, all the points occur in southern Italy.

Lines 163-166: Funnel traps are standard method for detecting aquatic species in general, and newts in particular - but were not used for these field studies... Why?

- The use of funnel traps is of course very useful but requests more fund and time for surveys; especially for the historical data gathering, VES and dip-netting techniques were used. Thus, to be consistent and standardize the methods, even all our recent data (on T. carnifex and other species) used for this paper were collected in the same way.

Lines 218-220: Considering your results, it would be interesting to compare these in terms of coverage per area. If the SACs have much larger area than the NDAs, than that could explain why they contain more localities (i.e., they don't necessarily cover more localities per area).

- Considering the great difference in surfaces between NDAs and Natura 2000 sites, we performed the analyses in terms of percentage (as reported in the corresponding figure, in the Y-axes of Figure 7 d and e). Indeed, we better specified this by adding a clarification in L 293.

Lines 251-253: Were funnel traps used in the "old" sampling period? If so, comparing these results with only visual searches of the recent surveys might not be very reliable.

- The “old” observations were also conducted by VES and dip-netting (see also our response above).

Lines 284-285, 328-329 and 362-363: Were all sites visited during the breeding season (unlikely, considering that visits were conducted until late summer)? If not, the reported "breeding activity" is not reliable. Provide more details.

- We actually report in the L 174-180 that “Sampling efforts were mainly focused from late winter to late summer, during the period of activity of the three species according to their phenology. The traditional meth-odologies for the detection and characterization of amphibians were used [59,60]. Briefly, we assessed the presence of amphibian species by visual encounter surveys for adults, egg clutches and larvae, by listening to the calls of adult males for B. pachypus, and by dip-nettings in aquatic sites. S. terdigitata were sampled by searching beneath logs, rocks, leaf–litter, trees' buttresses, and cavities.” We could also go deeper in explaining our field methods, but some other Reviewers and the Editor suggested us to even remove the whole paragraph, thus we prefer to maintain this level of detail.

Lines 447-451: This is an interesting topic for future research, but at this point I think it is pure speculation and should be removed.

- Thank you for this observation, we have deleted the period.

Lines 455-458: Actually it is the snow-line that matters for the rate of warming, not a specific altitude. Again, if you can't provide a reference to back your claim, I suggest you remove it.

- Thank you for this observation, we have deleted the sentence.